# Research on Risk Assessment and Contamination Monitoring of Potential Toxic Elements in Mining Soils

**DOI:** 10.3390/ijerph20043163

**Published:** 2023-02-10

**Authors:** Jie Yang, Yunlong Wang, Rui Zuo, Kunfeng Zhang, Chunxing Li, Quanwei Song, Xianyuan Du

**Affiliations:** 1State Key Laboratory of Petroleum Pollution Control, Beijing 102206, China; 2College of Water Sciences, Beijing Normal University, Beijing 100875, China; 3Engineering Research Center of Groundwater Pollution Control and Remediation, Ministry of Education, Beijing 100875, China

**Keywords:** potentially toxic elements, risk assessment, analytical hierarchy, potential ecological risk index, ordinary kriging, radial basis function

## Abstract

Potentially toxic element (PTE) contamination in soils has serious impacts on ecosystems. However, there is no consensus in the field of assessment and monitoring of contaminated sites in China. In this paper, a risk assessment and pollution monitoring method for PTEs was proposed and applied to a mining site containing As, Cd, Sb, Pb, Hg, Ni, Cr, V, Zn, Tl, and Cu. The comprehensive scoring method and analytical hierarchical process were used to screen the priority PTEs for monitoring. The potential ecological risk index method was used to calculate the risk index of monitoring point. The spatial distribution characteristics were determined using semi-variance analysis. The spatial distribution of PTEs was predicted using ordinary kriging (OK) and radial basis function (RBF). The results showed that the spatial distribution of As, Pd, and Cd are mainly influenced by natural factors, while Sb and RI are influenced by both natural and human factors. OK has higher spatial prediction accuracy for Sb and Pb, and RBF has higher prediction accuracy for As, Cd, and RI. The areas with high ecological risk and above are mainly distributed on both sides of the creek and road. The optimized long-term monitoring sites can achieve the monitoring of multiple PTEs.

## 1. Introduction

Potentially toxic elements (PTEs) in the soil are enriched through food crops and other means, which has an impact on human health and ecosystems [1,2,3,4,5,6,7]. Chinese soils are contaminated by PTEs to varying degrees, and the level of PTE contamination in southern China is generally higher than that in the north [8]. According to China’s soil contamination survey announcement in 2014, the total soil contamination rate in the country was 16.1%, and the number of inorganic contamination exceeding the standard accounted for 82.8% of all the contamination [9]. Human activities such as industrial projects and mining are the main cause of soil contamination exceeding standard levels [10]. More work is needed in China’s soil environmental monitoring to determine the criteria for priority contamination in the soil environment and to establish a monitoring network system. Therefore, monitoring and preventing PTEs contamination in the soil is urgent.

There are many multi-index evaluation methods applied in the field of PTE risk assessment. Different indicators, standards, and models provide different interpretations of the same data [11]. The method of fuzzy comprehensive evaluation is commonly used to grade the solid soil environmental quality, but it is mostly used to evaluate a single contamination, and composite contamination is rarely used [12,13]. If the index set is large, the weight vector and the fuzzy matrix do not match, resulting in the super blur phenomenon and poor resolution [14,15]. The analytical hierarchy process uses the principle of fuzzy mathematics to combine quantitative and qualitative methods to conduct hierarchical research on complex problems and is often used to solve multiobjective decision-making problems [16,17]. Yang et al. used the analytic hierarchy process and weighted average method to build a model to conduct real-time detection for soil PTE contamination in mining areas [18]. Yang et al. used the Analytic Hierarchy Process (AHP) to determine the weight of mixed entropy in the entropy-cloud model to quantify the problem of PTE assessment in agricultural fields [19]. 

At present, there are various methods for soil PTE risk assessment. Islam et al. used methods such as the contamination load index and potential ecological risk index to assess the soil environmental quality of 12 different land-use types in Bangladeshi cities [20]. Marrugo-Negrete et al. used four methods, the contamination factor method, enrichment factor method, geoaccumulation index, and risk assessment code, to evaluate PTE contamination in the soil of the Simi River agricultural irrigation area in northern Colombia [21]. Chen et al. modified and adjusted the potential ecological hazard coefficient and comprehensive ecological hazard index according to the type and amount of PTEs in the studied soil to improve the accuracy of the analysis results [14]. Wu et al. calculated the geoaccumulation index, pollution index, and potential ecological risk index to evaluate the soil contamination level in cities around electronic manufacturing plants [5]. He et al. used the comprehensive contamination index to evaluate the contamination characteristics of PTEs legally and quantitatively in soil in Daye City, Hubei [22]. The soil PTE contamination evaluation method essentially uses the calculation of numbers and formulas to reflect the contamination status of the study area. The commonly used methods for evaluating soil PTE contamination have limitations. The Nemeiro comprehensive pollution index method does not consider the difference in the toxicity of various contaminants, and it is difficult to express the qualitative change characteristics of contamination. The geoaccumulation index method focuses on single metals and does not consider the biological effects, the contamination contribution rate of each factor, or physical geographic differences [23]. The potential ecological risk index method comprehensively considers the environmental effects, ecological effects, and toxicology of PTEs, reduces the impact of regional differences, and reflects the corresponding relationship of PTEs to the ecological environment [24].

Determining the best sample size and monitoring location is crucial in studying the spatial changes in contaminations in the soil. The optimization study of monitoring points generally uses classical statistics, stratified sampling, geostatistical analysis, and kriging interpolation. Zhang et al. found that the number of sampling points determined by the classical geostatistical Cochran formula was insufficient, which affected the accuracy of the spatial prediction of soil organic carbon at the karst basin scale [25]. In the stratified sampling method, the proportion of each district is used for distribution, and the variability of the interval is ignored [26]. Compared with classical statistical methods, the geostatistical methods such as the Ordinary Kriging (OK) can maximize the use of various data provided by existing samples for spatial prediction and accuracy analysis. Maas et al. researched the PTE content in the surface layer of Annaba and its surrounding areas in Algeria and found that the grid layout combined with OK interpolation can accurately reflect the overall distribution of PTEs [27]. Qian Ding found that the precision of Radial Basis Function (RBF) interpolation for Pb and Zn is higher than that of OK interpolation [28]. The neural network based on the RBF has high accuracy for the distribution of PTEs in water [29].

Sweden and other developed countries have established soil monitoring networks, but the world has not yet reached an agreement on the number of monitoring points, placement locations, monitoring intervals, and other key issues [30]. In order to provide a basis for the risk assessment and contamination monitoring in contaminated sites, the research content of this paper includes the following three aspects: (1) screening priority monitoring indicators from PTEs and assessing the risk level of monitoring points. Through the comparative analysis of the comprehensive scoring method and analytic hierarchy process, PTEs with high scores were selected from the contaminated areas as priority monitoring indicators, and the potential ecological risk index method was used to determine the risk level of the monitoring points. (2) Spatial variability analysis and spatial distribution prediction of potential ecological risk indices. Semi-covariance function was used to analyze the spatial distribution characteristics of priority monitoring indicators. By comparing the accuracy of OK interpolation and RBF interpolation of each monitoring index, an appropriate interpolation method was selected to ensure the accuracy of the spatial distribution of potential ecological risk indices and identify high ecological risk areas. (3) The optimization of monitoring sites for soil PTEs. Based on the spatial distribution of potential ecological risks at various levels of monitoring indicators, the optimization principles were used to optimize the number and location of monitoring sites to reduce monitoring costs while achieving the goal of monitoring single-metal and multi-metal contamination levels. 

## 2. Materials and Methods

### 2.1. Sampling

The research area of this study is Lutang tailings pond located in Dachang Town, Nandan County, Hechi City, Guangxi Province, China. Soil samples were collected from the study area approximately 11 km^2^ to the southeast of the Lutang tailings reservoir in July 2017, and surface soil at a depth of 0–20 cm was collected. The specific distribution of sampling points is shown in Figure 1. A total of 135 soil samples were collected in the study area, including 68 dryland soil samples, 29 paddy soil samples, four woodland soil samples, and 34 other types of soil samples. 

### 2.2. Comprehensive Scoring Method

The comprehensive scoring method is a method of scoring and sorting PTEs based on the main screening indicators of PTE contamination in the study area, such as the pollution index, toxicity, bioaccumulation, and research basis, before the screening. The Pollution Index (PI) was calculated as follows:(1)$$PI=C_{j}/C_{b}$$
where $C_{j}$ is the concentration of the jth PTE in the soil sample, while $C_{b}$ is the background value of the target PTEs [31,32]. PI is represented by the symbol A. The background values of PTEs in Nandan County are shown in Table 1 [33].

The symbol B represents the acute toxicity index of rodent oral LD50 (mg/kg) [34]. C represents the carcinogenicity level of heavy metal compounds [35]. D is the bioaccumulation of heavy metals, expressed as the partition coefficient of octanol/water (lgKow) [36,37]. E and F represent whether heavy metals are priority pollutants in China and risk control projects in China, respectively [38,39].

When calculating the score of each PTE, a weight coefficient is introduced for weighted calculation. The weights coefficients and value intervals of each screening indicator are shown in Table 2. The specific formula is as follows:(2)$$T_{i}=A_{i}\times a_{i}+B_{i}\times b_{i}+C_{i}\times c_{i}+D_{i}\times d_{i}+E_{i}\times e_{i}+F_{i}\times f_{i}$$
where $A_{i}$, $B_{i}$, $C_{i}$, $D_{i}$, $E_{i}$ and $F_{i}$ are the screening indicators of the corresponding PTE i; $a_{i}$, $b_{i}$, $c_{i}$, $d_{i}$, $e_{i}$ and $f_{i}$ are the weight coefficients of each screening indicator [40]; and $T_{i}$. is the total score of PTE i.

As shown in Table 3, the monitoring interval was divided into three levels according to the total scores of each PTE $T_{i}$. The PTEs with scores in the first rank belong to the priority monitoring indicators; those in the second rank belong to the secondary monitoring indicators; and the remaining PTEs belong to the optional indicators.

### 2.3. Analytical Hierarchy Process

#### 2.3.1. Establishment of Analytical Hierarchy Process

The analytic hierarchy model used for screening PTE monitoring indicators in the tailing reservoir is a three-layer structure [41]. From top to bottom, it contains the target layer O, the standard layer C, and the protocol layer P. The element in the criterion level C includes the exceeding rate of metal contaminations in the study area, LD50, carcinogenicity, lgKow, whether it is a priority control pollutant in China, and whether it is part of China’s risk management and control projects, which are denoted as $C_{1}\ldots C_{6}$. The solution layer P is the soil weight metal screening index, which is denoted as $P_{1}\ldots P_{9}$ for nine metals: Cd, As, Sb, Pb, V, Hg, Zn, Tl, and Cu. The structure diagram of the analytic hierarchy process is shown in Figure 2.

#### 2.3.2. Judgment Function Structure

According to the specified scale method, the O → C layer determination matrix of PTE contamination, $C_{i}$ → P judges in the study area were established [42]. There were altogether 8 judgment matrices, and the matrix model is shown in the equation [43].
(3)$$A=\left[a_{ij}\right]=\left[\begin{array}{cccc} 1 & a_{12} & \cdots & a_{1n}\\ 1/a_{12} & 1 & \cdots & a_{2n}\\ \vdots & \vdots & \ddots & \vdots \\ 1/a_{1n} & 1/a_{2n} & \ldots & a_{nn} \end{array}\right]$$

As shown in Table 2, the weighting of the elements in the guidance tier is consistent with the weighting of the elements in the comprehensive scoring method.

#### 2.3.3. Calculation of the Matrix Feature Vector and Maximum Eigenvalue

The approximate calculation method for judging matrix feature vectors in the mining area soil is to first sum the judgment matrix and then obtain an average of each row to obtain a column vector. Each element of the vector is divided by the sum of all the elements of the column. Thus, the obtained column vector is feature vector W in the mining area soil [44]. For the characteristic vector of soil PTEs in the mining area, the algorithm of the minor metal maximum characteristic value $\lambda_{max}$ in the mining area is as follows:(4)$$\lambda_{max}=\sum_{i=1}^{n}\frac{{\left(A*W\right)}_{i}}{n*W_{i}}$$
where W is the characteristic vector of the soil PTE weight of the mining area, $W_{i}$ is the i element in the feature vector, A is the determination matrix of PTE contamination in the mining area, and n is the order of the determination matrix of PTE contamination in the mining area.

#### 2.3.4. Consistency Test

To confirm the judgment matrix, the average consistency index CI and consistency ratio CR are introduced. The calculation formula is as follows:(5)$$CI=\frac{\lambda_{max}-n}{n-1}$$

$\lambda_{max}$ is the maximum feature vector of the judgment matrix.
(6)$$CR=\frac{CI}{RI}$$
where RI is called the random consistency index and is given in Table 4. When CR < 0.1, it is considered that the matrix has satisfactory consistency. When CR > 0.1, the judgment matrix consistency is poor and needs to be re-estimated until the determination matrix consistency meets the requirements [16].

### 2.4. Potential Ecological Risk Index

The potential ecological risk index method is used to evaluate the degree of ecological risk caused by PTEs in soil and reflects the comprehensive environmental response of various PTEs by combining the content, quantity, and toxicity characteristics of PTEs in soil [20,45,46]. The specific formula is as follows:(7)$$C_{f}^{i}=C_{s}^{i}/C_{n}^{i}$$
(8)$$E_{r}^{i}=T_{r}^{i}*C_{f}^{i}$$
(9)$$RI=\sum_{i=1}^{n}E_{r}^{i}$$
where $C_{f}^{i}$ is the enrichment coefficient of PTE i; $C_{s}^{i}$ is the measured content of PTE i; $C_{n}^{i}$ is the reference value required for calculation; $E_{r}^{i}$ is the potential ecological risk coefficient of PTE i; $T_{r}^{i}$ is the toxicity coefficient of PTE i (As = 10, Pb = 5, Cd = 30, Sb = 20) [47,48]; RI is the comprehensive potential of the ecological risk indices for multiple PTE, the value of RI is related to the type and quantity of contaminations. In this study, RI was divided into five levels, as shown in Table 5.

### 2.5. Spatial Variability Analysis

The semi-covariance function was used to analyze the spatial variability and the factors influencing the spatial distribution. In this paper, a semi-covariance model was fitted to the potential ecological risk data of PTEs in the study area using Gs + 7.0 software. The semi-covariance model with the optimal spatial distribution of the data was selected based on the coefficient of determination and the residual value. The nugget gold value (Co) is caused by measurement error and intermittent variation smaller than the minimum sampling spacing, which is random variation; the abutment value (Co + C) reflects the degree of spatial variation generated by the synergistic effect of human and natural factors. The block gold coefficient (Co/(Co + C)) is a visual representation of the proportion of the role of natural and human factors among the factors influencing the degree of spatial variability. If Co/(Co + C) is less than 25%, it reflects that the spatial variation of the variable is mainly influenced by natural factors (rivers, etc.), and the variable has strong spatial correlation; when the ratio is in the range of 25% to 75%, the variable is moderately spatially correlated, and the spatial variation is jointly influenced by natural and human factors; when it is greater than 75%, the spatial correlation of the variable is weak, and the influence of human factors (village, roads, etc.) is large.

The range A_0_ represents the range of spatial autocorrelation of the PTEs’ potential ecological risk index in the study area [49]. The larger the range value, the stronger the homogeneity of the variable in the soil; the smaller the range value, the stronger the local variability of the variable and the worse the uniformity.

### 2.6. OK Interpolation

The OK interpolation method is a commonly used spatial method in geostatistics. The focus of the geostatistical analysis method is to use the existing sample point information to analyze the patterns (the variation function) of the research target (a certain variable) that changes with the spatial position and then to infer the attribute value of the unknown point [50,51]. Therefore, geostatistics is mainly used for the analysis and simulation of the spatial autocorrelation of variables. The expression of the variation function is shown below.
(10)$$r\left(h\right)=1/2N\left(h\right)\sum_{i=1}^{N\left(h\right)}{\left[Z\left(X_{i}\right)-Z\left(X_{i}+h\right)\right]}^{2}$$
where $r\left(h\right)$ is the semivariance function; h is the step length, which is the distance vector between two sample points; and $N\left(h\right)$ is the number of sample point pairs separated by *h*; $Z\left(X_{i}\right)$ and $Z\left(X_{i}+h\right)$, which represent the actual measured values of zone variable $Z\left(X\right)$ at points $X_{i}$ and $X_{i}+h$. If there is spatial autocorrelation, $r\left(h\right)$ increases with increasing h; when h reaches a certain distance H (that is, the range), $r\left(h\right)$ tends to stabilize, which is the base station value.

For the spatial distribution of PTE contamination in regional soils, current sampling point data are generally used for calculations and predictions through interpolation models. The accuracy of the OK method is mainly affected by the number of samples and spacing [52,53]. The kriging interpolation method estimates the data at any point $X_{0}$ based on the measured value of known points $X_{i}$ and assigns a weight coefficient $\lambda_{i}$ to the known value during estimation. The specific formula is as follows:(11)$$Z\left(X_{0}\right)=\sum_{i=1}^{n}\lambda_{i}Z(X_{i})$$

### 2.7. RBF Interpolation

Any function ϕ that satisfies the characteristic of $\varphi \left(x\right)=\varphi \left(\left\|x\right\|\right)$ is called a RBF. Take the known n different points $x_{1}$, $x_{2}$, ……, $x_{n}$, and the known function values $f\left(x_{1}\right)$, $f\left(x_{2}\right)$, ……, $f\left(x_{n}\right)$ as the training sample set [54]. Take the RBF $\varphi \left(x\right)$ to construct the surface model. The RBF interpolation model $S_{n}\left(x\right)$ has the form like this.
(12)$$S_{n}\left(x\right)=\sum_{i=1}^{n}\lambda_{i}\varphi \left(\left\|x-x_{i}\right\|\right)+p\left(x\right)$$
where $\left\|x-x_{i}\right\|$ represents the Euclidean distance between x and the center point $x_{i}$. $\lambda_{i}$ is a coefficient. $p\left(x\right)$ is a polynomial function. 

For a training data set with n samples, a linear system of equations with n unknown coefficients $\lambda_{i}$ can be obtained. The formula is as follows:(13)$$\left\{\begin{array}{c} \sum_{i=1}^{n}\lambda_{i}\varphi \left(\left\|x^{1}-x_{i}\right\|\right)=d^{1}\\ \sum_{i=1}^{n}\lambda_{i}\varphi \left(\left\|x^{2}-x_{i}\right\|\right)=d^{2}\\ \ldots \\ \sum_{i=1}^{n}\lambda_{i}\varphi \left(\left\|x^{n}-x_{i}\right\|\right)=d^{n} \end{array}\right.$$

The above equation can be changed to
(14)Φλ=D
(15)$$\lambda =\Phi^{-1}D$$
where Φ represents the element n∗n order matrix, λ and D represent the coefficient vector and the expected output vector, respectively. After obtaining the coefficient vector of the training set, the entire space mapping function value can be obtained according to the coordinates of the space [55]. This research used pycharm 2021 software to realize RBF spatial interpolation. RBF $\varphi \left(x\right)$ used in spatial interpolation include cubic, gaussian, inverse multiquadric, linear, multiquadric, quintic, and thin plate [56]. By adjusting different RBF, the accuracy of spatial interpolation can be improved.

### 2.8. Principles of Optimizing Monitoring Sites

On the basis of mastering the main types, contamination degrees, possible sources and spatial distribution characteristics of soil PTE contamination in the study area, the following optimization principles are proposed for the existing relatively intensive monitoring points [57].

#### 2.8.1. Representativeness

Due to different ecological risk levels, regional monitoring focuses are different. The representativeness of monitoring points is particularly important. (1) The more representative places, the more priority monitoring should be given. Monitoring sites in high-risk areas are more representative than those in low-risk areas. (2) The more PTEs with high risk levels contained in the monitoring site, the more representative the monitoring site is. (3) The monitoring points arranged on both sides of streams or roads are more representative. The soil PTE contamination in the study area was mainly caused by seepage, surface water hydraulics, and transportation. High-risk or contaminated areas tend to show trends along streams and roads, and changes in PTE content may be more pronounced than in other areas. (4) For low-risk areas, if there is no significant change in monitoring point information within one to two monitoring periods, it can be considered that the soil environment in this area is relatively stable, and no further monitoring can be considered.

#### 2.8.2. Comprehensiveness

The layout of different risk levels should be considered when optimizing the monitoring points. In addition to implementing key monitoring of high ecological risk areas, it is also necessary to pay attention to the changes in low ecological risk areas to ensure comprehensive monitoring of the entire study area [58]. Based on monitoring points in high-risk areas and supplemented by monitoring points in low-risk areas, the comprehensiveness of monitoring in the study area is guaranteed.

## 3. Results and Discussion

### 3.1. Screening of the Long-Term Monitoring Indicators

After monitoring the PTEs content in the soil, it was found that 11 PTEs in the soil exceeded standard levels. As shown in Figure 3, the contents of the Cr and Ni did not exceed the standard point. Therefore, these two PTEs were essentially harmless to the soil environmental quality of the study area. All other PTEs had varying levels of excess. Among them, the exceeding rates of As, V, Cd, and Sb were greater than 50%, indicating a greater risk of contamination.

In this study, we screened out monitoring indicators from nine metals using a comprehensive scoring method and analytic hierarchy process. The comprehensive scoring method calculates the scoring results of different screening indicators for each PTE in the study area, as shown in Table 6. The scoring order of the nine PTEs was Cd > As > Sb > Pb > Hg > V > Zn > Tl > Cu. For the scoring interval of this study, see the grading standard shown in Table 3. Among the nine PTEs, the priority monitoring indicator was Cd. The secondary monitoring indicators were As, Sb, Pb, and Hg. The rest were optional indicators. Therefore, the comprehensive scoring method identified Cd, As, Sb, Pb, and Hg as the main monitoring indicators for the study area.

In the calculation process of the analytic hierarchy process, six criterion-level elements were used to screen nine metals and check the consistency of the judgment matrix. The test results are shown in Table 7. The CR of the judgment matrix was less than 0.1. The consistency of the judgment matrices was satisfactory. The weight of each selected metal is shown in Table 8 below. According to the screening results of the analytic hierarchy process, the priority monitoring indicator was Cd, the secondary monitoring indicators were Sb, As, and Pb, and the rest were selective measurement indicators. The analytic hierarchy process screened Cd, Sb, As, Pb, and V as the main monitoring indicators in the study area.

The monitoring indicators screened out by the comprehensive scoring method and the hierarchical analysis method all contained Cd, Sb, As, and Pb. Therefore, the four abovementioned indicators were used as long-term monitoring indicators of soil PTEs to evaluate the ecological risk level.

### 3.2. Potential Ecological Risk Assessment

#### 3.2.1. Potential Ecological Risk Assessment of Monitoring Indicators

In this study, the soil environmental background in Nandan County was used as the reference standard, and the potential ecological risk index method was used to evaluate the ecological risk levels of the four detection indicators in the soil in the study area. The results are shown in Figure 4.

The potential ecological risk indices of As and Pb (EAs and EPb) were at the slightly high ecological risk level, and the percentages of points with high ecological risk and above were 17.77% and 13.33%, respectively. The point where the potential ecological risk index (ECd and ESb) of PTEs Cd and Sb exceeded 80% represents a medium high ecological risk or higher, indicating that the potential ecological risk of Cd and Sb in the study area was usually higher. The comprehensive potential ecological risk index (RI) of these four PTEs was almost all above the medium ecological risk level. A total of 27.41% of the points were at the high ecological risk level, and the proportions of the high and extremely high ecological risk levels were 11.11% and 19.26%, respectively.

#### 3.2.2. The Spatial Correlation Analysis of Potential Ecological Risks Index

The semivariance analysis was performed on the potential ecological risk index of soil PTEs monitoring indicators. The parameters of the semivariogram model of the potential ecological risk index are shown in Table 9. The nugget coefficient of As, Pd, and Cd is less than 25%, which has a strong spatial correlation, indicating that the spatial variability may have a greater correlation with natural factors such as river and soil parent material. The nugget coefficient of Sb and RI is between 25% and 75%, the spatial correlation is at a moderate level, and it is affected by the interaction of natural and man-made factors. 

The Variation of The PTEs and RI was higher than the interval of sampling points (100 m), indicating that the pollution uniformity of adjacent sampling points in the same risk area was good, and the difference was not obvious. Therefore, when optimizing the monitoring points, the adjacent points can be appropriately reduced in the same risk area to efficiently monitor the contaminated site.

### 3.3. Comparison between OK and RBF Interpolation

The data set had a total of 135 monitoring points, and the data set was randomly classified according to the ratio of eight: two and used as a training set and test set, respectively. Based on the training set, the OK was used for spatial interpolation, and the test set was used to verify the accuracy of the OK interpolation. The test set was brought into the model to obtain the prediction result. Finally, the error evaluation index was used to judge the accuracy of the interpolation method. 

Three error evaluation indicators, mean absolute error (MAE), mean square error (MSE), and coefficient of determination (R^2^), were used to evaluate the prediction performance of spatial interpolation. The smaller the MAE and MSE, the higher the prediction accuracy. The closer R^2^ is to one, the better the fitting effect is. The calculation formula is as follows:(16)$$MAE=\frac{1}{n}\sum_{i=1}^{n}\left|y_{i}-{\hat{y}}_{i}\right|$$
(17)$$MSE=\frac{1}{n}\sum_{i=1}^{n}{\left(y_{i}-\hat{y_{i}}\right)}^{2}$$
(18)$$R^{2}=1-\frac{\sum_{i}^{n}{\left(y_{i}-\hat{y_{i}}\right)}^{2}}{\sum_{i}^{n}{\left(y_{i}-\bar{y}\right)}^{2}}$$
where $\hat{y_{i}}$ is the predicted value, $y_{i}$ is the actual value, and $\bar{y}$ is the sample mean.

By comparing the data in the Table 10, the prediction of As and Cd by RBF with linear as the kernel function had different degrees of improvement than the accuracy of the OK. The range of MAE of As reduced by 1.94%. The range of MSE of As and Cd reduced by 20.0% and 10.5%. The R^2^ of As and Cd increased by 4.2% and 26.0%. The MAE and MSE of RI were reduced by 10.4% and 22.0%. The R^2^ of RI increased by 45.0%. Therefore, As and Cd use RBF, with linear as the kernel function, for spatial interpolation. RBF with thin plate as kernel function was used for spatial prediction of RI.

The prediction of Pb, Sb in the OK was more accurate than the RBF. Compared with RBF, the MAE of Pb and Sb using the OK interpolation were reduced by 4.7% and 6.5%, respectively. The range of MSE was reduced by 6.4% and 84.9%, and the R^2^ was improved by 21.9% and 131.0%. Thus, Pb and Sb were analyzed by spatial interpolation using the OK.

### 3.4. The Spatial Distribution of Potential Ecological Risks of Monitoring Indicators

In order to satisfy the conditions of OK interpolation, the data of ESb and EPb were reciprocally transformed to make the data nearly normal distribution.

According to the semivariance function model and related parameters, the OK was used to perform spatial interpolation analysis on the potential ecological risk index of the monitoring indicators, and the interpolation results intuitively showed the spatial distribution of each risk level. As shown in Figure 5, Pb and As were mostly at a slight ecological risk level. The ecological risk levels of Sb, Cd and RI were high and above in most areas. After a comparative analysis of five distribution charts, it was found that the locations of high-risk areas were relatively similar. The areas with high ecological risk and above were mainly distributed in villages and on both sides of Pingcun creek and highways. The spatial distribution of PTEs may be related to seepage, hydraulic action of surface water, and ore truck transport. The distribution characteristics of the abovementioned potential ecological risk indices of various metals are conducive to the optimization of contamination sites in the study area.

### 3.5. Optimize the Layout of Monitoring Points

According to the actual situation of the study area and the results of potential ecological risk assessment, the high-risk area was established as the key monitoring object, and the contamination path (leakage, surface water hydraulic effect, transportation, and other factors) and feasibility (funding and other factors) were considered. The monitoring principles of representativeness and comprehensiveness optimize the number and location of monitoring points for monitoring indicators of ecological risks. The optimization of the monitoring point of As can be used as an example.

From Figure 5 we can observe that the AS was spatially clustered. The high-risk area of As was located in the middle of the study area, and the risk index gradually decayed toward the southwest and northeast. On the whole, the risk of As in the northwest direction was high, and the risk in the southeast direction was low. To a certain extent, it satisfied the trend that the closer to the tailings area, the higher the concentration.

There were five risk levels for As in the study area. Although only three small areas are at the extremely high ecological risk level, considering the principle of prioritizing monitoring of high-risk areas, monitoring points must be set up in each small area. Points 46, 77, 129, and 130 remained. According to Spatial variability analysis, the spatial distribution of As is mainly affected by natural factors. Soil PTE pollution migrates through leakage, surface water hydraulics, and vehicle transportation. Therefore, the same risk level gives priority to close to natural rivers, which can reflect the location of natural elements. Because the range of As was much larger than the interval between adjacent points, it showed that there was no difference between adjacent points in the same risk area. Therefore, points 46 and 130 remained in the same area, 46 was kept, and point 130 was discarded. Monitoring points in high-risk areas were reserved for points 46, 77, and 129.

The very high ecological risk level areas were mainly distributed around the extremely high ecological risk level areas. Even though the monitoring points 48, 57, 58, and 59 were very close to the identified monitoring points 46 and 129, considering the principle of focusing on monitoring high-risk areas, two points were set up for monitoring in the very area. In the mining area, the spatial distribution of PTEs is mainly related to seepage, hydraulic action of surface water, and ore truck transportation. Therefore, the monitoring points of rivers, roads, and village attachments can more accurately monitor the risk changes of PTEs. Point 57 was closer to the Pingcun creek than point 58, and point 48 was closer to the Flat village and the country road than point 59. There were two smaller very high-ecological-risk-level areas around Sanhe village. Point 33 and point 133 were in the center of the area. Therefore, points 33, 48, 57, and 133 were reserved as monitoring points in very serious areas.

The third is the high ecological risk level. This area was mainly distributed in the north and west of Flat village, east of the Sanhe village and north of the Ratu village. Point 91 was in the center of the severe risk zone. Considering the comprehensive coverage of monitoring site distribution, point 119 was north of the study area and near the Tailing reservoir and the country road. Point 135 was the highest risk point in the southeast corner and is close to the country road. Points 119 and 135 were suitable as monitoring points.

When multiple monitoring points are deployed in high-risk areas, the layout of monitoring points in the medium and light ecological risk level areas should focus on the coverage of the entire study area. The center points 11 and 23 of the medium-ecological-risk-level area are the monitoring points in the south of the study area. Points 86 and 108 were close to the country road as monitoring points in the light-ecological-risk-level area.

Therefore, the number of optimization results for the monitoring program for As risk was 14, and the final point layout results are shown in Figure 6. The optimization process of monitoring points for As, Pb, Sb, and RI is similar to that of As. The results of the optimization of monitoring points are shown in Figure 6 below.

The point optimization results of various monitoring indicators showed that there were many similar monitoring points in the optimization results of the points. The fundamental reason is that there are multiple heavy metal compound contamination problems in the study area, which is conducive to reducing the monitoring workload to a certain extent. 

Combined with the optimized results of As, Pb, Cd, Sb, and RI monitoring points, if only a single metal content or changes in ecological risk need to be monitored, the optimization results of the corresponding metal monitoring points are shown in Figure 6. If monitoring the total contamination of five PTEs in the field, the optimized layout of RI can be used.

As shown in Figure 6, post-optimization layout of RI included 12 monitoring points. Points 33, 46, 77, 119, and 135 can monitor four PTEs indicators; points 48, 84 and 108 contain three types of PTE ecological risk monitoring. Points 3, 23, 62, and 111 can monitor one or two monitoring indicators. Although the point optimization scheme of RI can guarantee the monitoring effects of different indicators, there are some imperfections if the monitoring sites in RI are used as long-term monitoring sites for multiple metal contaminants in contaminated mining areas.

According to the statistics in Table 11, some points with low RI values but high ecological risk indexes of two PETs are easily overlooked, resulting in lack of accuracy and comprehensiveness of long-term monitoring. Although the RI of points 129 and 133 was not high, As and Pb could be monitored, and the ecological risk level of As and Pb was high and above. Therefore, 129 and 133 monitoring points were added on the basis of RI monitoring points, and a total of 14 monitoring points were finally obtained. On the basis of RI monitoring points, monitoring points 129 and 133 were increased and finally there were 14 monitoring points in total. 

Therefore, the 14 monitoring points in Figure 7 can be used as the final scheme for long-term monitoring of PTEs contamination risk in the study area. It can be found from Table 11 that multiple PTEs were monitored at each long-term monitoring site to achieve the purpose of monitoring combined pollution.

## 4. Conclusions

In this study, Pollution Index, Acute toxicity index of rodent oral, Carcinogenicity, Bioaccumulation, Priority monitoring of pollutants in China, and Risk control projects in China were selected as PTE screening indexes. Compared with single screening indexes, this screening method considers factors more comprehensively. The PTEs were ranked by the comprehensive scoring method and analytical hierarchical process. The order of the top four PTEs (Cd, Sb, As, and Pb) selected by the two methods is the same and they were used as priority monitoring indicators. When the weight coefficient of the index is the same, different screening methods have no obvious effect on the screening results. Using semi-variance analysis, we found that the spatial distribution characteristics of As, Pd, and Cd were mainly influenced by natural factors, while Sb and RI were influenced by both natural and anthropogenic factors. In the optimization of monitoring points, it is very important to determine the spatial variability analysis of monitoring elements. If the distribution of spatial characteristics is affected by natural factors, it is necessary to consider factors such as natural river soil parent material. If the spatial distribution characteristics are affected by human factors, it is necessary to consider factors such as villages and roads. By comparing the two interpolation methods of OK and RBF, OK had a higher spatial prediction accuracy for Sb and Pb, and RBF had a higher prediction accuracy for As, Cd, and RI. In the spatial interpolation prediction of some PTEs, the RBF interpolation method had higher prediction accuracy than the traditional OK interpolation method. Different elements need to select different interpolation methods to ensure the prediction accuracy. According to the spatial distribution map of PTEs, the areas with high ecological risk and above were mainly distributed on both sides of the creek and road. The spatial distribution of PTEs may be related to seepage, hydraulic action of surface water, and ore truck transportation. When making a single PTE contamination and composite PTE contamination monitoring point map, the principle of representativeness and comprehensiveness should be combined. The monitoring points in the composite PTE contamination monitoring point maps can monitor multiple PTEs and achieve the purpose of comprehensive monitoring. It is hoped that this study will provide some basis and reference for the assessment and monitoring of PTE-contaminated areas. 

## Figures and Tables

**Figure 1 ijerph-20-03163-f001:**
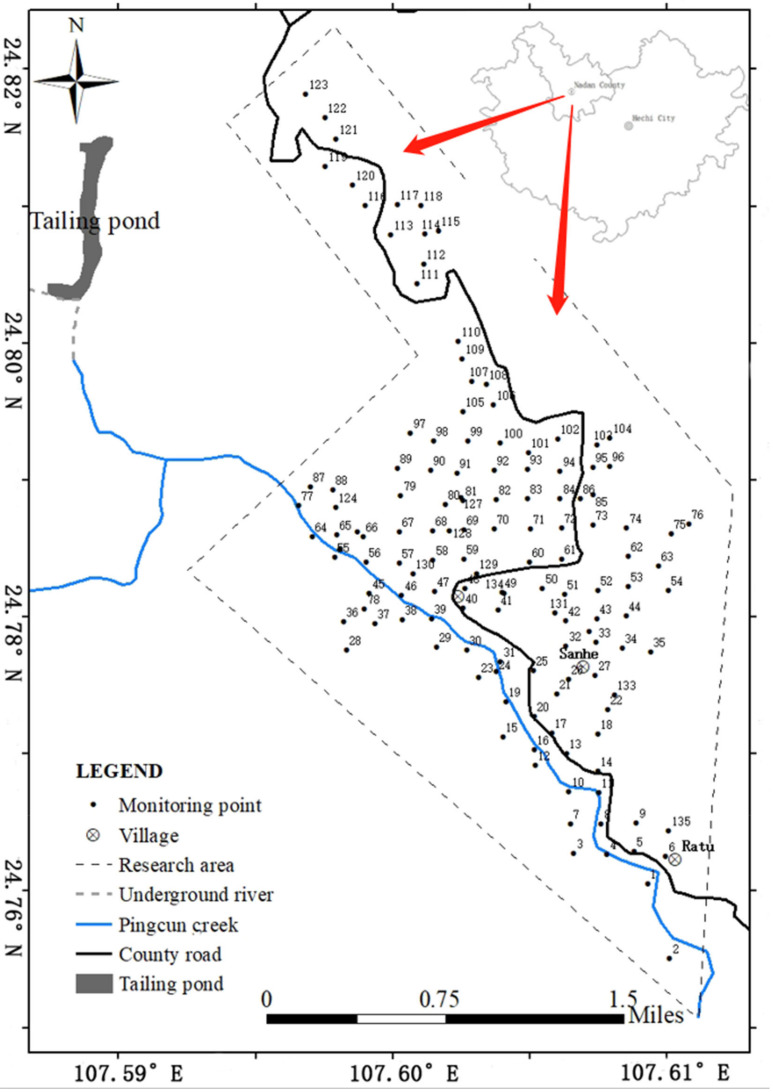
Location map of the study area in Dachang Town, Nandan County, Hechi City, Guangxi Province, China.

**Figure 2 ijerph-20-03163-f002:**
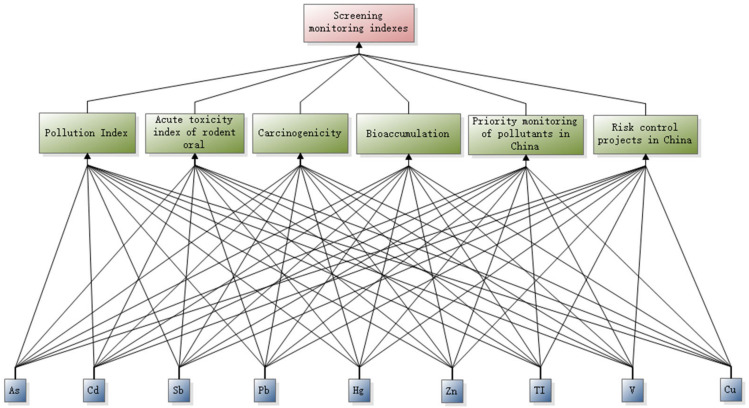
Hierarchical analysis structure diagram.

**Figure 3 ijerph-20-03163-f003:**
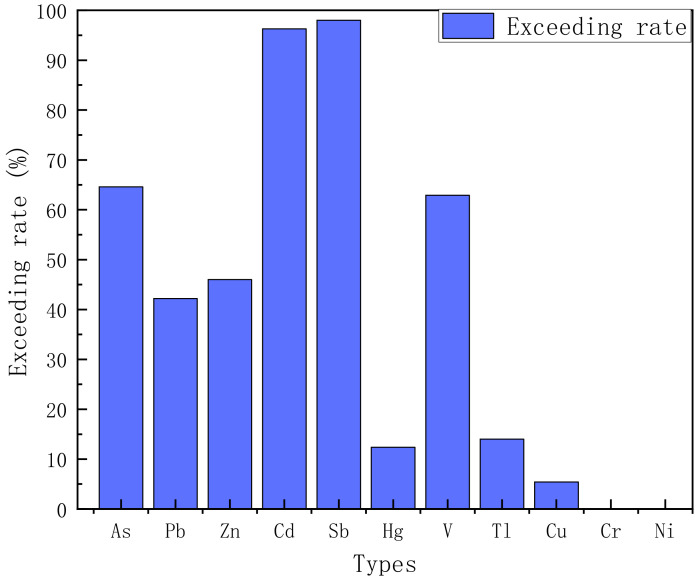
Excessive rate of PTE content.

**Figure 4 ijerph-20-03163-f004:**
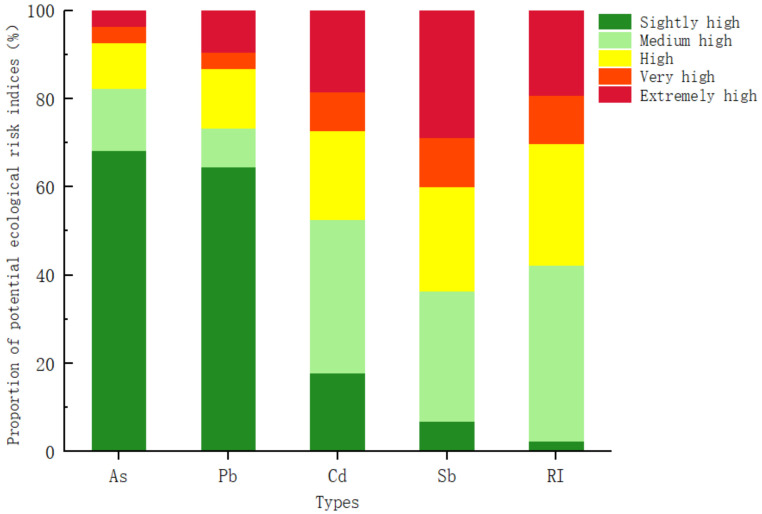
Percentage of potential ecological risk levels of monitoring indicators.

**Figure 5 ijerph-20-03163-f005:**
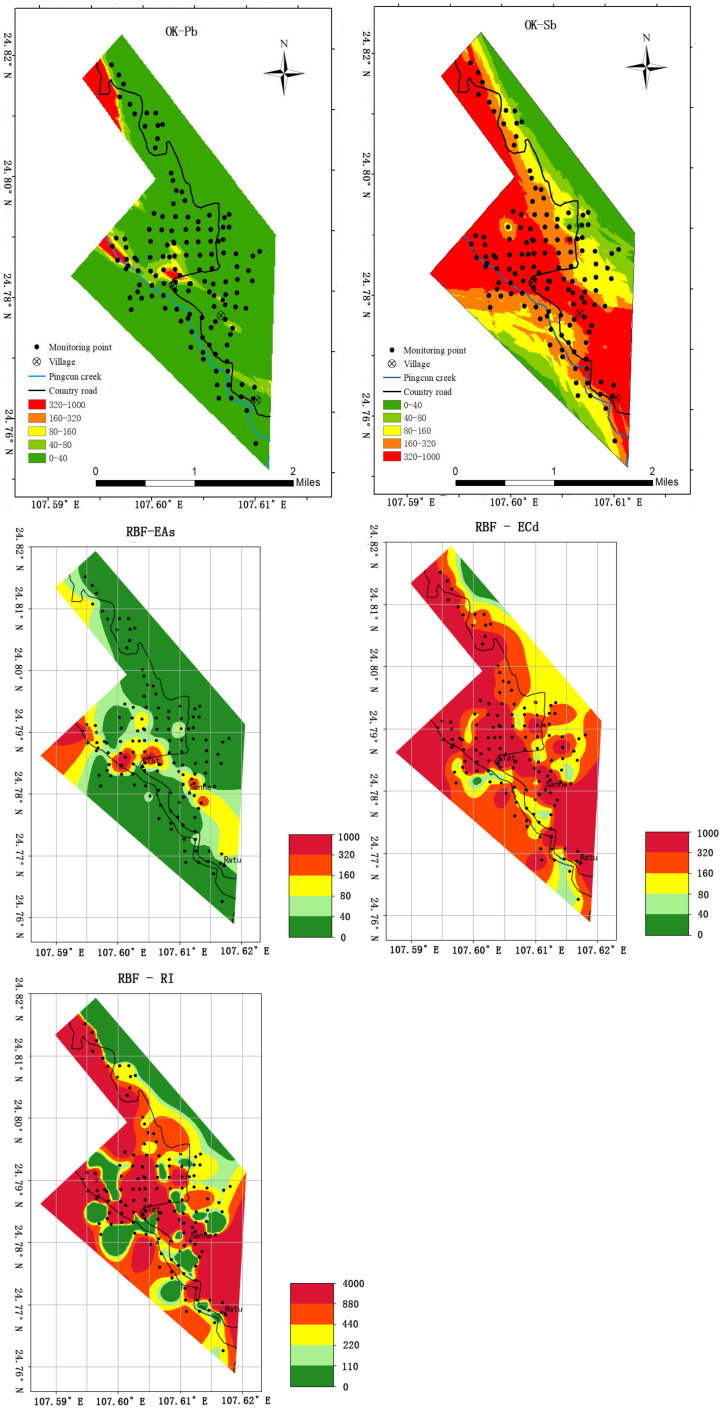
Interpolation results of potential ecological risks of soil As, Cd, Sb, Pb, and RI monitoring indicators.

**Figure 6 ijerph-20-03163-f006:**
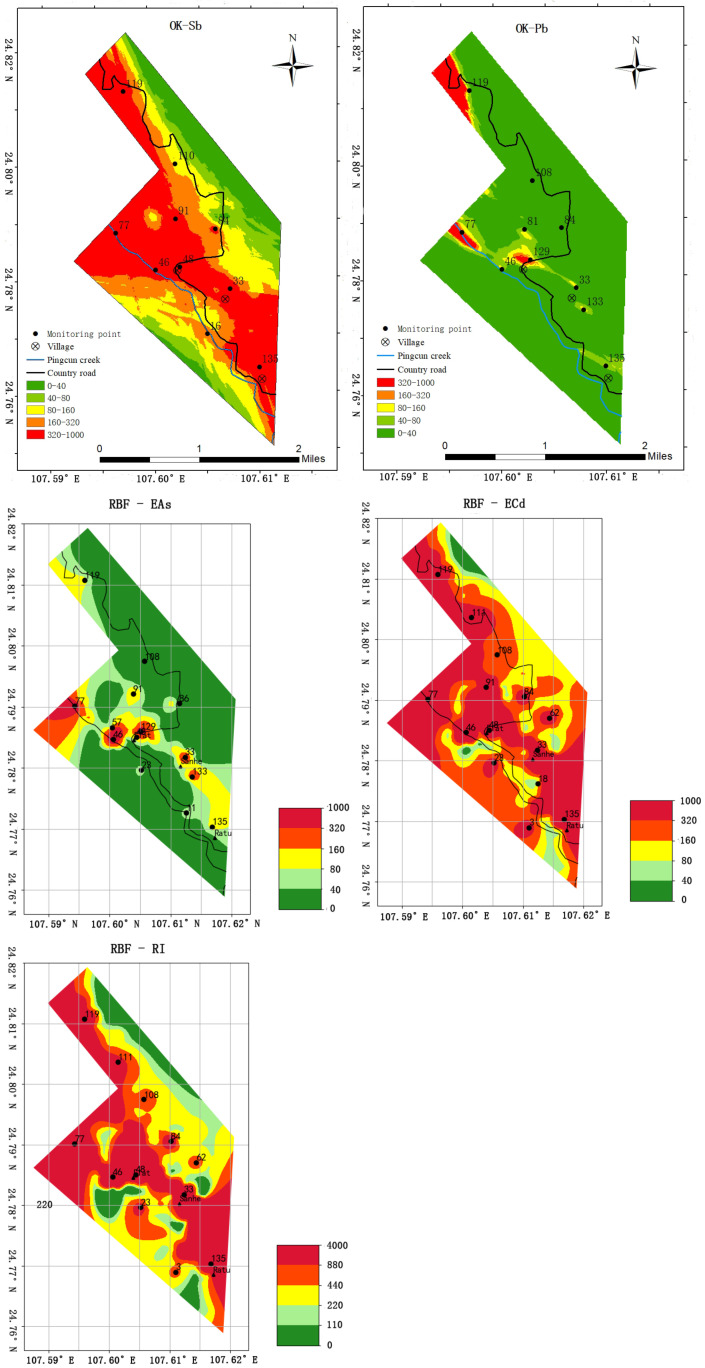
Post-optimization layout of Cd, Sb, As, Pb, and RI potential ecological risk monitoring points.

**Figure 7 ijerph-20-03163-f007:**
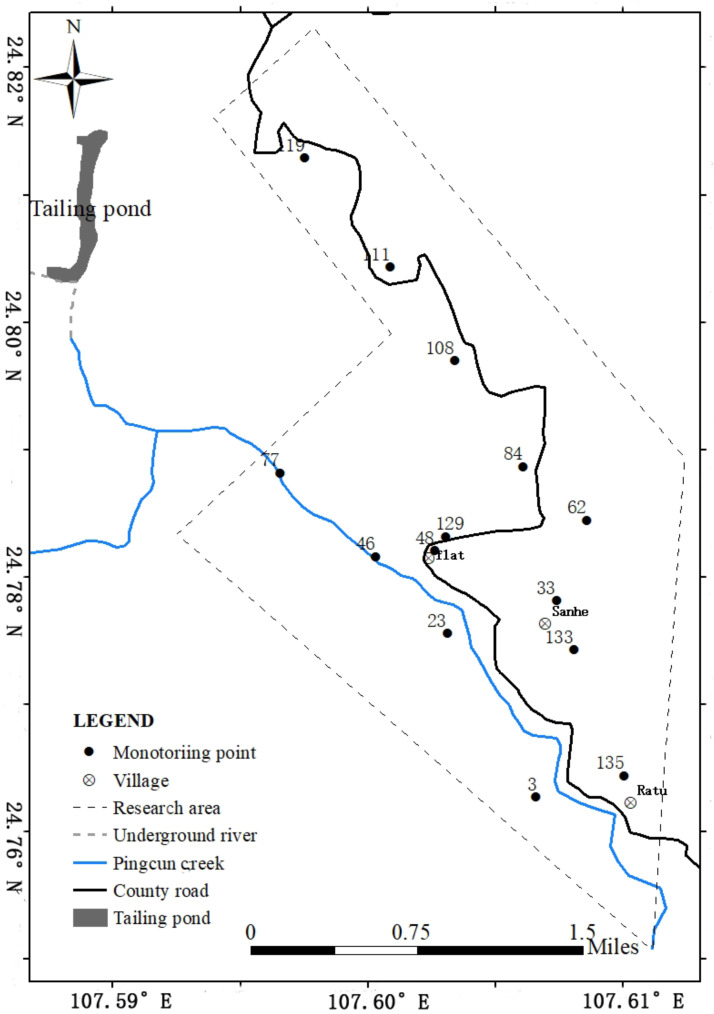
Distribution map of monitoring point optimization of the study area in Dachang Town, Nandan County, Hechi City, Guangxi Province, China.

**Table 1 ijerph-20-03163-t001:** The background values of potentially toxic elements (PTEs) in Nandan County.

Sort	As	Cr	Pb	V	Zn	Cd	Cu	Ni	Tl	Sb	Hg
Value (mg/kg)	21.6	75.0	28.7	142.8	82.9	0.1	23.2	26.9	0.7	2.9	0.3

**Table 2 ijerph-20-03163-t002:** Screening score of soil environmental quality monitoring indicators.

Label	Screening Indicators	Weights Coefficients	Points				
			1	2	3	4	5
A	Pollution Inx	25	1~1.2	1.2~1.4	1.4~1.6	1.6~1.8	>1.8
B	Acute toxicity index of rodent oral	3	no data	>5000	50~5000	5~50	<5
C	Carcinogenicity	3	no data	4	3	2	1
D	Bioaccumulation	5	no data	<3.5	3.5~4.2	>4.2	
E	Priority monitoring of pollutants in China	12	no				Yes
F	Risk control projects in China	6	no				Yes

**Table 3 ijerph-20-03163-t003:** Score scale of soil environmental quality monitoring indicators.

Priority of Indicator Monitoring	Total Score	Significance
Level 1	210~240	Priority monitoring
Level 2	160~210	Secondary monitoring
Level 3	60~160	Select monitoring

**Table 4 ijerph-20-03163-t004:** Average random consistency index.

Matrix Order	1	2	3	4	5	6	7	8	9
RI	0	0	0.58	0.90	1.21	1.24	1.32	1.41	1.45

**Table 5 ijerph-20-03163-t005:** Standards for the classification of potential ecological risks.

$E_{r}^{i}$	RI	Ecological Risk Level
$E_{r}^{i}$ < 40	RI < 110	slight
40 ≤ $E_{r}^{i}$ < 80	110 ≤ RI < 220	medium
80 ≤ $E_{r}^{i}$ < 160	220 ≤ RI < 440	high
160 ≤ $E_{r}^{i}$ < 320	440 ≤ RI < 880	Very high
$E_{r}^{i}$ ≥ 320	RI ≥ 880	Extremely high

**Table 6 ijerph-20-03163-t006:** Soil monitoring index score results.

Index	Pollution Index	Acute Toxicity Index of Rodent Oral	Carcinogenicity	Bioaccumulation	Priority Monitoring of Pollutants in China	Risk Control Projects in China	Total
Cd	5	3	3	2	3	5	219
As	4	3	4	4	3	5	207
Sb	5	2	5	4	1	1	184
Pb	3	3	4	1	3	5	167
Hg	2	4	1	4	3	5	151
V	4	1	1	3	1	1	139
Zn	3	3	1	2	1	5	139
Tl	2	4	5	1	3	1	124
Cu	1	3	1	1	3	5	108

**Table 7 ijerph-20-03163-t007:** Consistency check index of various levels.

Arrangement	$\lambda_{max}$	CR
$O-C_{i}$	6.0083	0.0013
$C_{1}-P_{i}$	9.1423	0.0122
$C_{2}-P_{i}$	9.1778	0.0152
$C_{3}-P_{i}$	9.2095	0.0179
$C_{4}-P_{i}$	9.0853	0.0073
$C_{5}-P_{i}$	9.0000	0.0000
$C_{6}-P_{i}$	9.1132	0.0097

**Table 8 ijerph-20-03163-t008:** Weight of each metal through the analytic hierarchy process.

Sort	Cd	Sb	As	Pb	V	Hg	Zn	TI	Cu
Weight	0.1622	0.1585	0.1451	0.1117	0.1007	0.0933	0.0899	0.0782	0.0605

**Table 9 ijerph-20-03163-t009:** Potential ecological risk index semivariance function model and related parameters of soil PTE monitoring indicators.

Project	MODEL	Nugget Value Co	Abutment Value Co + C	Nugget Coefficient Co/Co + C (%)	Variation A_0_ (m)	Decisive Factor r^2^	Residual RSS
EAs	Spherical	2.41 × 10^−4^	1.92 × 10^−3^	12.54	1470	0.817	6.82 × 10^−7^
EPb	Spherical	6.99 × 10^−4^	3.08 × 10^−3^	22.71	748	0.417	3.85 × 10^−6^
ECd	index	0.022	0.208	10.54	411	0.354	5.86 × 10^−3^
ESb	Spherical	0.149	0.440	33.87	891	0.563	4.10 × 10^−2^
RI	Spherical	0.102	0.257	39.50	681	0.438	1.33 × 10^−2^

**Table 10 ijerph-20-03163-t010:** The comparison of prediction errors of PTE spatial interpolation between the Ordinary Kriging (OK) and Radial Basis Function (RBF).

Heavy Metal	OK			RBF			
	MAE	MSE	R^2^	MAE	MSE	R^2^	Kernel Function
As	0.0155	0.0005	0.6838	0.0152	0.0004	0.7125	linear
Cd	0.2610	0.1127	0.2880	0.2705	0.1009	0.3629	linear
Pb	0.0325	0.0018	0.3415	0.0341	0.0019	0.2801	Inverse quadratic
Sb	0.1320	0.0349	0.9080	0.3775	0.2325	0.3931	linear
RI	0.3694	0.2344	0.3281	0.3316	0.1917	0.4504	thin plate

**Table 11 ijerph-20-03163-t011:** Statistical table of the number of monitoring indicators for different optimization points.

Serial Number	Sample Number	EAs	EPb	ECd	ESb	RI
1	3			√		√
2	11	√				
3	16				√	
4	18			√		
5	23	√		√		√
6	33	√	√	√	√	√
7	46	√	√	√	√	√
8	48	√		√	√	√
9	57	√				
10	62			√		√
11	77	√	√	√	√	√
12	81		√			
13	84		√	√	√	√
14	86	√				
15	91	√		√	√	
17	108	√	√	√		√
18	110				√	
19	111			√		√
20	119	√	√	√	√	√
21	129	√	√			
22	133	√	√			
23	135	√	√	√	√	√

## Data Availability

The data presented in this study are available on request from the corresponding author. The data are not publicly available due to privacy.

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
