# Peer review of "Research on Risk Assessment and Contamination Monitoring of Potential Toxic Elements in Mining Soils"

_ijerph, 2023, doi:10.3390/ijerph20043163_

Round 1
Reviewer 1 Report
The contents of this manuscirpts are sound, but the structure and language thorought the whole manuscript, can be improved for a better paper. This includes the modification of several nouns (e.g., soil heavy metal pollution evaluation methods, the heavy metal monitoring indicators, fuzzy comprehensive evaluation methods), definition of acronys of OK and RBF both in Abstract and main manuscrit, change 4 to four (for a number of less than 10), choice of words, coherency among paragraphs and between sentences (e.g., Lines 111-112 are abrupt from above sentences, and this is also true for Lines 115-116).
Author Response
Dear Reviewer:
Thank you very much for your suggestions for revisions to the manuscript. We have read your suggestions carefully and found that there are indeed problems with our manuscript. In response to your comments, I have revised and clarified each of them as follows.
1、This includes the modification of several nouns (e.g., soil heavy metal pollution evaluation methods, the heavy metal monitoring indicators, fuzzy comprehensive evaluation methods)
Answer:(1) “Commonly used soil heavy metal pollution evaluation methods have certain limitations.” was changed to “The commonly used methods for evaluating soil PTEs contamination have limitations”;
(2)“The comprehensive scoring method and the analytical hierarchy process were used to compare the heavy metal monitoring indicators” was changed to “The comprehensive scoring method and analytical hierarchical process were used to screen the priority PTEs for monitoring.”
(3)”There are fuzzy comprehensive evaluation methods for soil environmental testing indicators, but fuzzy comprehensive evaluation methods are mostly used to evaluate a single pollutant, and composite pollution is rarely used” was changed to ” The method of fuzzy comprehensive evaluation is commonly used to grade the solid soil environmental quality, but it is mostly used to evaluate a single contaminatant, and composite contamination is rarely used”
2、definition of acronys of OK and RBF both in Abstract and main manuscript
Answer:The full names of OK(ordinary kriging) and RBF(radial basis function) have been added in the abstract and in the first appearance of the text, while the abbreviated OK and RBF are used only after the first appearance of the full names.
3、change 4 to four (for a number of less than 10)
Answer:We double-checked the article and converted all similar numbers (less than 10) into alphabetic form
4、choice of words
Answer: Since the study included metals and metalloids, we changed heavy metal to potentially toxic elements for the sake of terminological accuracy. In addition, we replaced the inappropriate nouns in the article.
5、coherency among paragraphs and between sentences (e.g., Lines 111-112 are abrupt from above sentences, and this is also true for Lines 115-116).
Answer: The question you asked is accurate. We deleted the sentences in Lines 111-112 and illustrated the meaning expressed in that sentence in line 101 to make the meaning of the whole article flow more smoothly . In addition, we have changed Lines 115-116 to “The research area of this study is Lutang tailings pond located in Dachang Town, Nandan County, Hechi City, Guangxi Province, China.”
The above is an item-by-item response to your questions. If you have other suggestions, please feel free to contact me. Finally, thank you again for your sincere suggestions.

Reviewer 2 Report
The manuscript is very well written, organised, and can be accepted for publication. Few of my comments needs to be addressed:
1. Author can provide a suitable graphical abstract as to signify what has been done by them in one go.
2. Minor spell checks/ spacing issues needs to be eradicated.
3. Some of the recent valuable papers such as: Journal of Molecular liquids,362,119752, Reactive and Functional Polymer 175(2022), 105261; Academic press, Elsevier, (2021), 449-460 ISBN 9780128240588, https://doi.org/10.1016/B978-0-12-824058-8.00037-2 can be included in the introduction section.
5. Conclusion of study needs to be improved.
Author Response
Dear Reviewer:
Thank you very much for your suggestions for revisions to the manuscript. We have read your suggestions carefully and found that there are indeed problems with our manuscript. In response to your comments, I have revised and clarified each of them as follows.
1、Author can provide a suitable graphical abstract as to signify what has been done by them in one go.
Answer:We produce graphic summaries to better present the content of the article.

2、Minor spell checks/ spacing issues needs to be eradicated.
Answer:We have revised some of the nouns (e.g., soil heavy metal pollution evaluation methods, the heavy metal monitoring indicators, fuzzy comprehensive evaluation methods et al) and the structure of the sentences to make the meaning more accurate and the structure more clear.
3、Some of the recent valuable papers such as: Journal of Molecular liquids,362,119752, (https://doi.org/10.1016/j.molliq.2022.119752)Reactive and Functional Polymer 175(2022), 105261; Academic press, Elsevier, (2021), 449-460 ISBN 9780128240588, https://doi.org/10.1016/B978-0-12-824058-8.00037-2 can be included in the introduction section.
Answer:We have carefully studied the three articles, the contents of which are closely related to the context of this study. It has been added to the article as a reference. The references are numbered 2,4,7。
4、 Conclusion of study needs to be improved.
Answer:Your question accurately points out the article's shortcomings. We have reorganized the conclusion section to be a clearer and more concise point of view. The details are as follows:
In this study, Pollution Index, Acute toxicity index of rodent oral, Carcinogenicity, Bioaccumulation, Priority monitoring of pollutants in China, and Risk control projects in China were selected as PTEs screening indexes. Compared with single screening indexes, this screening method considers factors more comprehensively. The PTEs were ranked by the comprehensive scoring method and analytical hierarchical process. The order of the top four PTEs (Cd, Sb, As and Pb) selected by the two methods is the same and were used as priority monitoring indicators. When the weight coefficient of the index is the same, different screening methods have no obvious effect on the screening results. Using semi-variance analysis, we found that the spatial distribution characteristics of As, Pd, and Cd were mainly influenced by natural factors, while Sb and RI were influenced by both natural and anthropogenic factors. In the optimization of monitoring points, it is very important to determine the spatial variability analysis of monitoring elements. If the distribution of spatial characteristics is affected by natural factors, it is necessary to consider factors such as natural river soil parent material. If the spatial distribution characteristics are affected by human factors, it is necessary to consider factors such as villages and roads. By comparing the two interpolation methods of OK and RBF, OK has a higher spatial prediction accuracy for Sb and Pb, and RBF has a higher prediction accuracy for As, Cd, and RI. In the spatial interpolation prediction of some PTEs, the RBF interpolation method has higher prediction accuracy than the traditional OK interpolation method. Different elements need to select different interpolation methods to ensure the prediction accuracy. According to the spatial distribution map of PTEs. The areas with high ecological risk and above are mainly distributed on both sides of greek and road. The spatial distribution of PTEs may be related to seepage, hydraulic action of surface water and ore truck transportation. When making a single PTEs contamination and composite PTEs contamination monitoring point map, the principle of representativeness and comprehensiveness should be combined. The monitoring points in the composite PTEs contamination monitoring point map can monitor multiple PTEs and achieve the purpose of comprehensive monitoring. It is hoped that this study will provide some basis and reference for the assessment and monitoring of PTEs contaminated areas.
The above is an item-by-item response to your questions. If you have other suggestions, please feel free to contact me. Finally, thank you again for your sincere suggestions.

Author Response
Dear Reviewer:
Thank you very much for your suggestions for revisions to the manuscript. We have read your suggestions carefully and found that there are indeed problems with our manuscript. In response to your comments, I have revised and clarified each of them as follows.
1、GPS position and location name of the study area should include in Figure 1 and Figure 7.
Answer:We have redrawn Figure 1 and Figure 7. The latitude and longitude coordinates were added to the original. In addition, the location of the study area is noted in detail.

Figure 1. Location map of the study area in Dachang Town, Nandan County, Hechi City, Guangxi Province, China.

Figure 7. Distribution map of monitoring point optimization of the study area in Dachang Town, Nandan County, Hechi City, Guangxi Province, China.
2、Line No 138, should include reference for background values of heavy metals.
Answer:The background values of heavy metals reference has been added and the reference number is 34.(34. Guangxi Institute of Environmental Science. Soil background value research method and background value of Guangxi soil.; Guangxi Science and Technology Press:Nanning, China,1992; pp.216-223)
3、Should be cleared with unity about Weights, is it weight or number of samples in Table 2?
Answer:This place is indeed inaccurately worded. We have changed the weights to weight coefficients. The correct one is the weight coefficients not the number of samples.
4、Line No 156 – 161 sentence structure should improve
Answer:The original sentence structure has problems leading to unclear expression. The modified sentences are as follows:
As shown in Figure 3, the monitoring interval was divided into three levels according to the total scores of each PTEs . The PTEs with scores in the first rank belong to the priority monitoring indicators; those in the second rank belong to the secondary monitoring indicators; and the remaining PTEs belong to the optional indicators.
5、Language and sentence structure should improve.
Answer:After listening to your suggestions, we modified the noun (e.g., soil heavy metal pollution evaluation methods, the heavy metal monitoring indicators, fuzzy comprehensive evaluation methods et al) and sentence structure of the whole article to make the logic and content clearer.

Reviewer 4 Report
The authors should rewrite the conclusions so that they focus on the main findings. What they contribute to the existing knowledge? The implications and benefits.
Author Response
Dear Reviewer:
Thank you very much for your suggestions for revisions to the manuscript. We have read your suggestions carefully and found that there are indeed problems with our manuscript. In response to your comments, I have revised and clarified each of them as follows.
1、 The authors should rewrite the conclusions so that they focus on the main findings. What they contribute to the existing knowledge? The implications and benefits.
Answer:Your question accurately points out the article's shortcomings. We have reorganized the conclusion section to be a clearer and more concise point of view. The details are as follows:
In this study, Pollution Index, Acute toxicity index of rodent oral, Carcinogenicity, Bioaccumulation, Priority monitoring of pollutants in China, and Risk control projects in China were selected as PTEs screening indexes. Compared with single screening indexes, this screening method considers factors more comprehensively. The PTEs were ranked by the comprehensive scoring method and analytical hierarchical process. The order of the top four PTEs (Cd, Sb, As and Pb) selected by the two methods is the same and were used as priority monitoring indicators. When the weight coefficient of the index is the same, different screening methods have no obvious effect on the screening results. Using semi-variance analysis, we found that the spatial distribution characteristics of As, Pd, and Cd were mainly influenced by natural factors, while Sb and RI were influenced by both natural and anthropogenic factors. In the optimization of monitoring points, it is very important to determine the spatial variability analysis of monitoring elements. If the distribution of spatial characteristics is affected by natural factors, it is necessary to consider factors such as natural river soil parent material. If the spatial distribution characteristics are affected by human factors, it is necessary to consider factors such as villages and roads. By comparing the two interpolation methods of OK and RBF, OK has a higher spatial prediction accuracy for Sb and Pb, and RBF has a higher prediction accuracy for As, Cd, and RI. In the spatial interpolation prediction of some PTEs, the RBF interpolation method has higher prediction accuracy than the traditional OK interpolation method. Different elements need to select different interpolation methods to ensure the prediction accuracy. According to the spatial distribution map of PTEs. The areas with high ecological risk and above are mainly distributed on both sides of greek and road. The spatial distribution of PTEs may be related to seepage, hydraulic action of surface water and ore truck transportation. When making a single PTEs contamination and composite PTEs contamination monitoring point map, the principle of representativeness and comprehensiveness should be combined. The monitoring points in the composite PTEs contamination monitoring point map can monitor multiple PTEs and achieve the purpose of comprehensive monitoring. It is hoped that this study will provide some basis and reference for the assessment and monitoring of PTEs contaminated areas.

Reviewer 5 Report
The manuscript presents the optimization of monitoring plan for metal polluted soils. Please find below some comments that have to be addressed before the manuscript could be considered for publication.
The title does not reflect the content of the manuscript. Please revise.
Please revise the abstract as there are several typos and grammar errors (some examples: L 10-11 but it is no consensus has been reached; L13 screening monitoring, L13 delate contaminated sites). Also improve the abstract to better reflect the manuscript content
Carefully check the spelling and grammar through the whole manuscript.
As you analyzed both metals and metalloids, I would suggest to use either potentially toxic elements (PTEs) or toxic elements.
Keywords: remove “site optimization”.
L52. Please revise
L97-99 revise the objective of the study to be clearer. Rephrase ”heavy metal monitoring indicators” for examples with “screening the metal content and assessment …”
L110 do you refer to monitoring sites or monitoring parameters?
L136 use either metal or element not both
L138-140 Add reference for data in Table 1. Add also the bkg values for Cr. If Mn is not studied you can remove-it from table 1.
L141-149. Please revise and clarify
Table 2 revise the name of screening indicators
L307 remove heavy metals, the mentioning of the element is enough
Table 6. Define each parameter (A-F) in the table caption
L332 replace strong with high when referring to ecological risk
Table 9 define abutment and nugget
L380 revise
L414 replace ‘heavy metal As’ with ‘As’. Revise through the whole manuscript
L423 you refer to Pingcun creek? If yes please revise.
L429-430 revise
L434, L474-475 please clarify
Conclusions-please revise and shorten the abstract.
Author Response
Dear reviewer:
Thank you very much for your suggestions for revisions to the manuscript. We have read your suggestions carefully and found that there are indeed problems with our manuscript. In response to your comments, I have revised and clarified each of them as follows.
1、The title does not reflect the content of the manuscript. Please revise.
Answer: We modify the title of the article. Change “Research on the optimization plan of heavy metal polluted soil monitoring in mining areas” to “Research on risk assessment and contamination monitoring of potential toxic elements in mining soils”
2、Please revise the abstract as there are several typos and grammar errors (some examples: L 10-11 but it is no consensus has been reached; L13 screening monitoring, L13 delate contaminated sites). Also improve the abstract to better reflect the manuscript content.
Answer: We have modified the grammar and sentence structure errors you mentioned. At the same time, the content of the abstract was reorganized. The latest abstract is as follows:
Potentially toxic elements (PTEs) contamination in soils has serious impacts on human health and ecosystems. However, there is no consensus in the field of assessment and monitoring of contaminated sites in China. In this paper, a risk assessment and pollution monitoring method for PTEs is proposed and applied to the mining site containing As, Cd, Sb, Pb, Hg, Ni, Cr, V, Zn, Tl and Cu. The comprehensive scoring method and analytical hierarchical process were used to screen the priority PTEs for monitoring. The potential ecological risk index method was used to calculate the risk index of monitoring point. The spatial distribution characteristics were deter-mined using semi-variance analysis. The spatial distribution of PTEs was predicted using ordi-nary kriging (OK) and radial basis function (RBF). The results show that the spatial distribution of As, Pd, and Cd are mainly influenced by natural factors, while Sb and RI are influenced by both natural and human factors. OK has higher spatial prediction accuracy for Sb and Pb, and RBF has higher prediction accuracy for As, Cd, and RI. The areas with high ecological risk and above are mainly distributed on both sides of greek and road. The optimized long-term monitor-ing sites achieve the monitoring of multiple PTEs.
- Carefully check the spelling and grammar through the whole manuscript.
Answer: According to your requirements, we adjust the inappropriate nouns (eg: soil heavy metal pollution evaluation methods, the heavy metal monitoring indicators, fuzzy comprehensive evaluation methods), wrong grammar and sentence structure so that the content of the article can be clearly expressed.
4、As you analyzed both metals and metalloids, I would suggest to use either potentially toxic elements (PTEs) or toxic elements.
Answer: According to your requirements, we study a number of related literature. We found that the noun potentially toxic elements (PTEs) is more suitable for this study than metal. Finally, we changed the heavy metals in the text to potentially toxic elements (PTEs).
5、Keywords: remove “site optimization”.
Answer: Combined with the research content of the article and your requirements, we modify the keywords to make it more accurate and comprehensive. The modified keywords are as follows:
Keywords: potentially toxic elements; risk assessment; analytical hierarchy; potential ecological risk index; ordinary kriging; radial basis function
6、L52. Please revise
Answer: “A comprehensive evaluation system that comprehensively considers soil selenium content, soil nutrients and soil environmental quality was established through the analytic hierarchy process, and screened out contamination indicators to provide a basis for the development of selenium-rich agriculture in the Boshan area” was changed to “Yang et al. used the Analytic Hierarchy Process (AHP) to determine the weight of mixed entropy in the entropy-cloud model to quantify the problem of PTEs assessment in agricultural fields”
7、L97-99 revise the objective of the study to be clearer. Rephrase ”heavy metal monitoring indicators” for examples with “screening the metal content and assessment …”
Answer: Combined with your suggestions, we will specify three aspects. The three aspects are as follows: (1) Screening priority monitoring indicators from PTEs and assessing the risk level of monitoring points. (2) Spatial variability analysis and spatial distribution prediction of potential ecological risk indices. (3) The optimization of monitoring points for soil PTEs
8、L110 do you refer to monitoring sites or monitoring parameters?
Answer: This is indeed a clear mistake, we have changed the monitoring indicators into monitoring sites.
9、L136 use either metal or element not both
Answer: We have modified the full text to PTEs.
10、L138-140 Add reference for data in Table 1. Add also the bkg values for Cr. If Mn is not studied you can remove-it from table 1.
Answer: We have added the references of Table1 to the article. The literature number is 34. In addition, we remove Mn and add the background value of Gr.
11、L141-149. Please revise and clarify
Answer: We redefined the screening indicators of PTEs and added relevant references. For example, carcinogenicity was analyzed according to the data published on the official website of Iternational Agency for Research on Cancer (IARC).
12、Table 2 revise the name of screening indicators
Answer: We modified the name of the screening index. Change LD50 to Acute toxicity index of rodent oral; Change lgKow to Bioaccumulation; Chang “Whether it is China's optimal control pollutant” to “Priority monitoring of pollutants in China”; Chang “Whether it is China's risk management and control projects” to “Risk control projects in China”
13、L307 remove heavy metals, the mentioning of the element is enough
Answer: We have deleted the heavy metal and modified it in the full text.
14、Table 6. Define each parameter (A-F) in the table caption
Answer: We replace the symbols in Table 6 with the names of the screening indicators. A(Pollution Index), B(Acute toxicity index of rodent oral), C(Carcinogenicity), D(Bioaccumulation), E(Priority monitoring of pollutants in China), F(Risk control projects in China).
15、L332 replace strong with high when referring to ecological risk
Answer: We replace strong with high in the whole article. At the same time, the notes in the figure 4 were modified.
16、Table 9 define abutment and nugget
Answer: We added a new 2.5. section and explained the Abutment value, Nugget coefficient, Nugget value and Variation in detail.
17、L380 revise
Answer: “The prediction of RI by RBF with thin plate as the kernel function is more accurate than OK.” This sentence is indeed unreasonable, we have deleted it from the article.
18、L414 replace ‘heavy metal As’ with ‘As’. Revise through the whole manuscript
Answer: We have deleted the heavy metal and modified it in the full text.
19、L423 you refer to Pingcun creek? If yes please revise. L429-430 revise; L434,
Answer: You may have questions about why to keep monitoring points near the river. The original expression is indeed more abrupt and not easy to understand. So we modified it in the article. At the same time, we analyze the spatial characteristics and possible migration paths of As in the article. According to Spatial variability analysis, the block gold coefficient of As is 12.54 % less than 25. The spatial distribution of As is mainly affected by natural factors (In chapter 2.5, there is a detailed explanation of each exclusive noun). In section 2.8.1, the third point of the principle of representativeness refers to soil PTEs pollution migrates through leakage, surface water hydraulics, and vehicle transportation. PTEs in places such as roads and rivers vary greatly and are the key monitoring places. Therefore, the same risk level gives priority to close to natural rivers, which can reflect the location of natural elements. Because the variation of As is much larger than the interval between adjacent points, it shows that there is no difference between adjacent points in the same risk area. Therefore, points 46 and 130 in the same area, keep 46, and discard point 130. For the question of retaining 47 and 48 monitoring points (L429-430,L434), it is similar to the point 46.
We also made a detailed description in the article to ensure that the content is complete and the structure is rigorous.
21、L474-475 please clarify
Answer: According to the statistics in Table 11, some points with low RI values but high ecological risk indexes of two PETs are easily overlooked, resulting in lack of accuracy and comprehensiveness of long-term monitoring. Although the RI of points 129 and 133 is not high, As and Pb can be monitored, and the ecological risk level of As and Pb are high and above. Therefore, 129 and 133 monitoring points were added on the basis of RI monitoring points, and a total of 14 monitoring points were finally obtained. We hope our explanation will remove your doubts. At the same time, We also explained in detail in the article.
22、Conclusions-please revise and shorten the abstract.
Answer: According to your requirements, we have streamlined the abstracts and conclusions. The details are as follows.
Abstracts: Potentially toxic elements (PTEs) contamination in soils has serious impacts on ecosystems. However, there is no consensus in the field of assessment and monitoring of contaminated sites in China. In this paper, a risk assessment and pollution monitoring method for PTEs is proposed and applied to the mining site containing As, Cd, Sb, Pb, Hg, Ni, Cr, V, Zn, Tl and Cu. The comprehensive scoring method and analytical hierarchical process were used to screen the priority PTEs for monitoring. The potential ecological risk index method was used to calculate the risk index of monitoring point. The spatial distribution characteristics were determined using semi-variance analysis. The spatial distribution of PTEs was predicted using ordinary kriging (OK) and radial basis function (RBF). The results show that the spatial distribution of As, Pd, and Cd are mainly influenced by natural factors, while Sb and RI are influenced by both natural and human factors. OK has higher spatial prediction accuracy for Sb and Pb, and RBF has higher prediction accuracy for As, Cd, and RI. The areas with high ecological risk and above are mainly distributed on both sides of greek and road. The optimized long-term monitoring sites achieve the monitoring of multiple PTEs.
Conclusion: In this study, Pollution Index, Acute toxicity index of rodent oral, Carcinogenicity, Bioaccumulation, Priority monitoring of pollutants in China, and Risk control projects in China were selected as PTEs screening indexes. Compared with single screening indexes, this screening method considers factors more comprehensively. The PTEs were ranked by the comprehensive scoring method and analytical hierarchical process. The order of the top four PTEs (Cd, Sb, As and Pb) selected by the two methods is the same and were used as priority monitoring indicators. When the weight coefficient of the index is the same, different screening methods have no obvious effect on the screening results. Using semi-variance analysis, we found that the spatial distribution characteristics of As, Pd, and Cd were mainly influenced by natural factors, while Sb and RI were influenced by both natural and anthropogenic factors. In the optimization of monitoring points, it is very important to determine the spatial variability analysis of monitoring elements. If the distribution of spatial characteristics is affected by natural factors, it is necessary to consider factors such as natural river soil parent material. If the spatial distribution characteristics are affected by human factors, it is necessary to consider factors such as villages and roads. By comparing the two interpolation methods of OK and RBF, OK has a higher spatial prediction accuracy for Sb and Pb, and RBF has a higher prediction accuracy for As, Cd, and RI. In the spatial interpolation prediction of some PTEs, the RBF interpolation method has higher prediction accuracy than the traditional OK interpolation method. Different elements need to select different interpolation methods to ensure the prediction accuracy. According to the spatial distribution map of PTEs. The areas with high ecological risk and above are mainly distributed on both sides of greek and road. The spatial distribution of PTEs may be related to seepage, hydraulic action of surface water and ore truck transportation. When making a single PTEs contamination and composite PTEs contamination monitoring point map, the principle of representativeness and comprehensiveness should be combined. The monitoring points in the composite PTEs contamination monitoring point map can monitor multiple PTEs and achieve the purpose of comprehensive monitoring. It is hoped that this study will provide some basis and reference for the assessment and monitoring of PTEs contaminated areas.

Round 2
Reviewer 3 Report
Figure 5, Figure 6, Figure 7 should present clearly (high quality).
Author Response
Dear Reviewer:
Thank you very much for your suggestions for revisions to the manuscript. We have read your suggestions carefully and found that there are indeed problems with our manuscript. In response to your comments, I have revised and clarified each of them as follows.
- Figure 5, Figure 6, Figure 7 should present clearly (high quality).
Answer:According to your revisions, we redraw and modify Figure 5,6,7. The specific graphics are as follows:

Figure 5. interpolation results of potential ecological risks of soil As, Cd, Sb, Pb, and RI monitoring indicators.

Figure 6. Post-optimization layout of Cd, Sb, As, Pb, and RI potential ecological risk monitoring points.
Figure 7 Upload failure, you can view from the attachment.

Reviewer 5 Report
The authors improved their manuscript. There are still some typos: Line 1034 there are some words beginning with capitalized letters in the sentence. The name of the river is Pingcun greek or Pingcun creek?
Author Response
Dear reviewer:
Thank you very much for your second review of my manuscript. We have read your suggestions carefully and found that there are indeed problems with our manuscript. In response to your comments, I have revised and clarified each of them as follows.
1、The authors improved their manuscript. There are still some typos: Line 1034 there are some words beginning with capitalized letters in the sentence. The name of the river is Pingcun greek or Pingcun creek?
Answer: We changed Pingcun Greek into Pingcun creek in the Figure 1,5,6,7 and text.
The above is the response to your questions. If you have other suggestions, please feel free to contact me. Finally, thank you again for your sincere suggestions.